# The Effectiveness of a Very Low-Volume Compared to High-Volume Laxative in Colon Capsule Endoscopy

**DOI:** 10.3390/diagnostics13010018

**Published:** 2022-12-21

**Authors:** Benedicte Schelde-Olesen, Artur Nemeth, Gabriele Wurm Johansson, Ulrik Deding, Thomas Bjørsum-Meyer, Henrik Thorlacius, Gunnar Baatrup, Anastasios Koulaouzidis, Ervin Toth

**Affiliations:** 1Department of Surgery, Odense University Hospital, 5700 Svendborg, Denmark; 2Department of Clinical Research, University of Southern Denmark (SDU), 5230 Odense, Denmark; 3Department of Gastroenterology, Skåne University Hospital, 205 02 Malmö, Sweden; 4Department of Surgery, Skåne University Hospital, 205 02 Malmö, Sweden; 5Department of Medicine, Odense University Hospital, 5700 Svendborg, Denmark; 6Department of Social Medicine and Public Health, Pomeranian Medical University, 70-204 Szczecin, Poland

**Keywords:** capsule endoscopy, bowel preparation, bowel cleansing quality, completion rate

## Abstract

Colon capsule endoscopy (CCE) is a promising modality for colonic investigations, but completion rates (CR) and adequate cleansing rates (ACR) must be improved to meet established standards for optical colonoscopy. Improvements should be made with patient acceptability in mind. We aimed to compare a very low-volume polyethylene glycol (PEG) laxative to a conventional high-volume laxative. We carried out a single-center retrospective comparative cohort study including patients referred for CCE. One hundred and sixty-six patients were included in the final analysis, with eighty-three patients in each group. We found a CR and ACR of 77% and 67% in the high-volume group and 72% and 75% in the very low-volume group, respectively. In the high-volume group, 54% had complete transit and adequate cleansing, whereas this was the case for 63% in the very low-volume group. No statistically significant difference in CR, ACR, or a combination of the two was found. A very low-volume bowel preparation regimen was non-inferior to a high-volume regimen before CCE in terms of CR and ACR.

## 1. Introduction

Colon capsule endoscopy (CCE) is a promising modality for lower gastrointestinal (GI) investigations in both clinical routine and screening [1,2,3,4,5,6,7,8,9,10]. Furthermore, the double-headed camera capsules are being applied for panenteric investigations, with promising results [11,12,13,14,15,16]. To achieve viable wider CCE adoption, challenges regarding completion rates (CR) and adequate cleanliness rates (ACR) must be handled [17,18]. CR and ACR should be improved to meet the standards for optical colonoscopy (OC) from the European Society of GI Endoscopy (ESGE). ESGE recommends both CR and ACR ≥ 90% [19]. Recently, a meta-analysis of preparation regimens for CCE confirmed that CR and ACR were suboptimal [20]. Although polyethylene glycol (PEG)-based laxatives and sodium phosphate (NaP)-based boosters were the most commonly used, the combination did not lead to higher CR or ACR. A meta-analysis exploring patient preferences for CCE and OC found that procedural adverse events with CCE were rare, the tolerability was high for both CCE and OC, and patient preferences for the two procedures did not differ significantly even though the tolerability for CCE was rated higher than for OC [21]. The results of the meta-analysis are based on preference for the entire pathway and patients undergoing only one of the examinations were excluded. The lack of polypectomy and biopsy capabilities in CCE and the risk of a second round of bowel preparation may sway the vote in favor of OC for patients with pathology where therapeutic procedures are needed. Bowel cleansing is known to be an obstacle to patient compliance with endoscopic procedures. A very low-volume laxative was developed to pursue a more tolerable cleansing regimen. So far, this regimen was only studied for bowel preparation before colonoscopy [22,23,24,25,26]. Therefore, this study aimed to investigate the CR, ACR, and diagnostic yield (DY) of a very low-volume PEG-based laxative compared to a conventional high-volume laxative.

## 2. Materials and Methods

We conducted a retrospective comparative cohort study. All consecutive patients referred to clinical CCE at Skåne University Hospital, Malmö, in the 3 years from July 2019 to June 2022, were included. According to the clinical protocol, patients under the age of 18 and patients with swallowing difficulties or previous intestinal resection were not eligible for CCE. We used the PillCam^®^ Crohn’s capsule (PCC) (Medtronic, Minneapolis, MN, USA) for all investigations. The PCC is used in clinical practice at Skåne University Hospital to enable panintestinal investigation, as a large proportion of the population referred to endoscopy at the department are patients with known or suspected inflammatory bowel disease (IBD). All patients were treated as day cases. The patient ingested only the first dose of laxative at home without supervision. A nurse with experience in CCE supervised the remaining steps in the procedure (Table 1). Patients stayed at the clinic until capsule excretion or until 4 pm.

### 2.1. Bowel Preparation

The same protocol was followed for all patients except for the laxatives used (Table 1). The high-volume group received 4 L of PEG solution (Laxabon^®^, Karo Pharma AB, Stockholm, Sweden). In contrast, the very low-volume group received 1 L PEG solution with ascorbic acid (PEG-ASC) (Plenvu^®^, Norgine, Amsterdam, The Netherlands) and 2 L of additional clear liquids. Both are taken in split doses. Sodium phosphate (Phosphoral, Recordati AB, Kista, Sweden) was used as a booster for all. If the capsule was not passed from the stomach to the small bowel (SB) within two hours from ingestion, it was endoscopically delivered to the duodenum.

### 2.2. CCE Evaluation

All capsule videos were analyzed using PillCam™ reader software v9 (Given Imaging Inc., Duluth, GA, USA). A gastroenterologist with extensive experience in SB capsule endoscopy (CE) and CCE evaluated cleanliness on the 4-point Leighton–Rex scale [27]. The grades excellent and good were considered adequate, while the grades fair and poor were considered inadequate (Figure 1). The reader was blinded to the bowel preparation regimen used. Complete CCE was defined as visualization of the hemorrhoidal plexus or an excreted capsule. The transit times were calculated for the stomach, SB, and colon separately by PillCam™ reader software.

### 2.3. Statistical Analysis

Baseline characteristics were compared between the high-volume and very low-volume groups, including age, sex, BMI, prior colonoscopy, and CCE indication. CE performance (completion, cleansing grade, retention, and transit times) and DY were additionally compared between the groups. DY and performance measures (except CR) were stratified by colon and SB. Transit time was additionally reported for the stomach. As the continuous variables were not normally distributed, the non-parametric Wilcoxon rank sum test was used for comparison. Categorical variables were compared using the chi^2^ test. Ordinal logistic regression models were conducted testing predictors of prolonged transit times (divided by 33rd and 67th percentiles), including age, sex, BMI, laxative, and indication for CE in the models. Data management and statistical analysis were conducted in SAS 9.4 (SAS, Gary, NC, USA).

## 3. Results

### Study Population

During the study period, 170 eligible patients underwent CCE. Four patients were excluded due to capsule retention in the esophagus or interruption in the bowel preparation phase (Figure 2). The number of excluded patients and the reason for exclusion was identical in the two groups. Patients were between 19 and 87 years old and were referred based on varying indications. The most common referral indication was known or suspected IBD, followed by overt and occult GI bleeding. The remaining 20% of patients were referred due to abdominal pain/diverticulitis, colon cancer screening, and weight loss (Table 2).

We found a CR and colonic ACR of 77% and 67%, respectively, in the high-volume group and 72% and 75% in the very low-volume group. In the high-volume group, 54% had complete transit with adequate colon cleansing, whereas this was the case for 63% in the very low-volume group (Figure 3). The differences between the two groups regarding CR, colonic ACR, or a combination of the two were not statistically significant. The only statistically significant difference was the stomach transit time, longer in the very low-volume group (Table 3). The DY was similar in the two groups, 59% in the high-volume group and 66% in the very low-volume group (Table 4). Ordinal logistic regression models found that increased age increased the total transit time whereas very low volume PEG increased the stomach transit time, adjusted for BMI, sex, and indication for CE. No significant adverse events requiring medical intervention or hospitalization occurred in either of the study groups.

## 4. Discussion

Our results show that a very low-volume bowel cleansing regimen before CCE performs equal to a high-volume regimen. We found no difference between the two groups except for a longer stomach transit time in the very low-volume group. As per the clinical routine at Skåne University Hospital, a gastroscopy was performed to deliver the capsule to the small bowel if it did not progress from the stomach within two hours. Considering this, stomach transit times could potentially be longer, with possible segmental delay or even stomach retention as a result.

The recent European introduction of a very low-volume PEG bowel preparation for OC prompted comparisons to established low- and high-volume regimens. So far, the very low-volume preparation shows to be at least non-inferior to both 2 L PEG-ASC and 4 L PEG solutions, with favorable patient experience, tolerability, and high adherence [22,23,24,25,26].

A recently published Swedish study on bowel preparation before OC showed that vomiting was more frequent in patients who received the very low-volume regimen [22]. Clinically, this could be a symptom of gastric transit delay, which is in line with our findings. The high dose of ascorbic acid in the very low-volume laxative (7.54 g) [28] is likely a large contributor to the GI symptoms. High oral doses of ascorbic acid is associated with GI symptoms such as abdominal pain, bloating and transient osmotic diarrhea [29]. The hyper-osmolality created by the ascorbic acid delivered in the low volume of the laxative solution facilitates bowel cleansing [30].

The results from the Swedish study showed that despite the vomiting associated with the very low-volume regimen, the overall experience reported by the patients was better than in the high-volume group [22]. Similar results for tolerability were found in an Italian study comparing high and very low-volume PEG regimens [23].

Our study showed a tendency towards higher small bowel and colon ACRs in the very low-volume group despite arithmetically lower CR. Considering this, choosing a different booster than the NaP-based booster could be a part of the solution. The results from a recent meta-analysis on bowel preparation for CCE suggest that adding the booster gastrografin could improve the CR [20]. The prokinetic prucalopride is also a promising option, as shown in a recent Danish study, where both CR and ACR were significantly improved with the addition of prucalopride to the booster regimen [31].

The 1 L PEG-ASC laxative is classified as a very low-volume laxative with only 1 L of active substance compared to 4 L in the high-volume regimens. Even though the laxative volume is 1 L, the total liquid volume is only 1 L less than in the high-volume regimens when adding the additional liquid recommended. However, it might be easier for the patients to accept the intake of additional fluids instead of the laxative solution, leading to the higher tolerability reported for 1 L-PEG-ASC compared to high-volume laxatives [22,23]. The additional liquid intake seems essential in improving CR. A study found that a water intake of ≥12 mL per minute during the CCE was an independent predictor for a complete CCE [32].

In some countries, CCE is already a part of the clinical practice. In March 2021, NHS England announced that CCE would be offered instead of OC to certain patient cohorts with low-risk colonic symptoms. Across NHS health boards in Scotland, CCE was implemented as a supplement to OC [33]. It was introduced as part of the ScotCap study, which found that CCE is both safe and well tolerated and that the proportion of patients in need of OC can be reduced. However, the Scottish team did find a need for improvement as the CR and ACR in their cohort were below 80%. Real-world data from Denmark and France confirm that outside the scope of academic trials, improvement is needed to increase the reliability of CCE [34,35]. Less than half of the French sample CCEs were considered optimal, i.e., complete with adequate bowel cleansing and in the Danish introduction of CCE in patients with incomplete colonoscopy, a CR of 60% and ACR of 54% was achieved, while only 40% had complete CCE with adequate bowel cleansing.

The present study is, to our knowledge, the first one examining the very low-volume bowel preparation in CCE specifically. Often it is difficult to report on patient adherence to the bowel preparation, as the data are often self-reported, introducing the risk of recall bias. In our study, dedicated endoscopy nurses supervised the bowel preparation (except for the first dose of laxatives), thereby controlling patients’ adherence to the protocol. CCE faces the challenge of delivering results on ACR comparable to OC, which is currently the gold standard for investigating colonic symptoms. However, the interest in CCE and the use of the modality is growing.

We acknowledge some limitations to our study. Because of the retrospective design, information on patient tolerability is not available. As the patients’ experience is very important when it comes to bowel cleansing, such data would heighten the conclusions that we can draw from this study. The design where study groups are not included in parallel introduces the risk of external factors affecting the results. No significant changes were made during the 3-year study period, and staff as well as protocols were consistent, minimizing the risk of biased results.

To investigate the possibilities for very low-volume bowel preparation in CCE further, studies in a prospective multicenter setting are needed.

## 5. Conclusions

A very low-volume bowel preparation regimen performed equal to a high-volume regimen before CCE in terms of CR and ACR. The markedly lower volume of active substance could be an advantage in regard to patient acceptability. To improve CR and possibly ACR further, studies on different boosters in addition to the laxative are warranted.

## Figures and Tables

**Figure 1 diagnostics-13-00018-f001:**
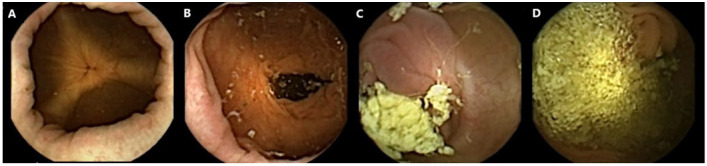
Examples of colon capsule video frames graded according to the Leighton–Rex scale. (**A**) Excellent. (**B**) Good. (**C**) Fair. (**D**) Poor.

**Figure 2 diagnostics-13-00018-f002:**
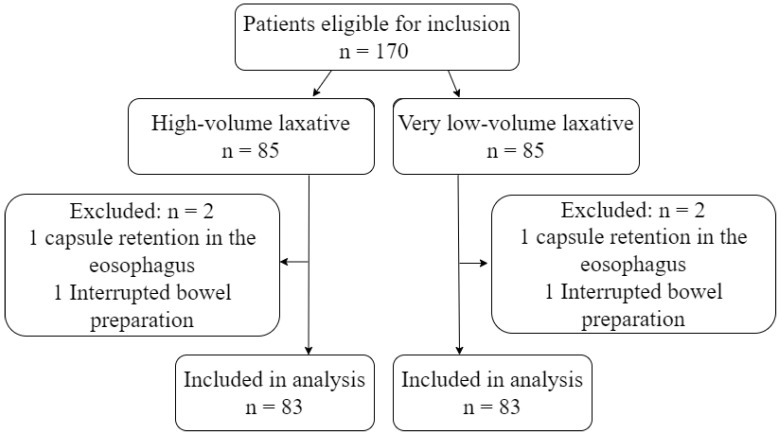
Flowchart.

**Figure 3 diagnostics-13-00018-f003:**
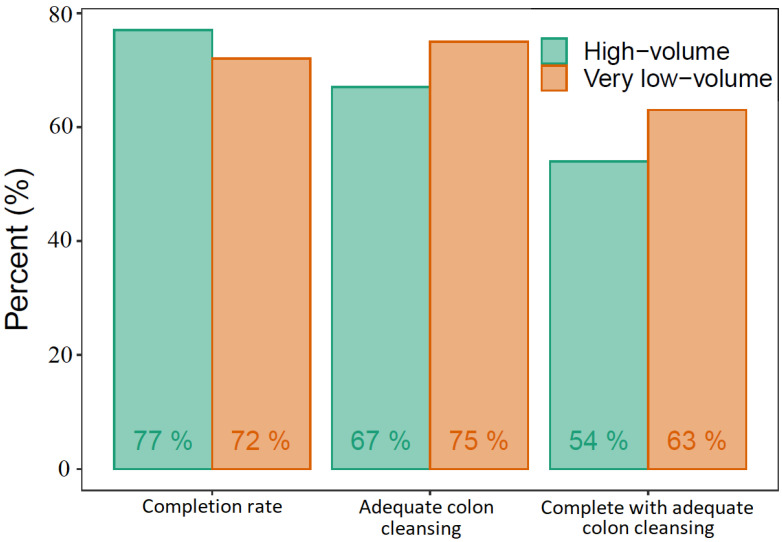
Capsule endoscopy performance.

**Table 1 diagnostics-13-00018-t001:** Bowel preparation regimens.

Time	High-Volume Regimen	Very Low-Volume Regimen
One day prior to capsule endoscopy	Low fiber diet. Light breakfast and lunch followed by clear liquid diet.	
	3 L PEG at 6–8 pm	0.5 L PEG-ASC+1 L water at 6–7 pm
45–75 min prior to capsule ingestion	200 mg Simethicone	
	1 L PEG	0.5 L PEG-ASC+1 L water
Capsule ingestion	150 mL water + 200 mg Simethicone	
First hour after capsule ingestion	Chewing gum *	
One hour after capsule ingestion	Metoclopramide 10 mg *	
1st boost:After capsule entry into the small bowel	30 mL Sodium phosphate^®^ + 1 L clear liquids	
2nd boost: 3 h after 1st boost **	15 mL Sodium phosphate^®^ + 1 L clear liquids	
3rd boost: 2 h after 2nd boost **	20 mg bisacodyl suppository	

* Optional only if capsule in stomach. ** Only if the capsule is not excreted. Steps, where regimens differ, are marked in italics. PEG: polyethylene glycol, PEG-ASC: polyethylene glycol + ascorbic acid.

**Table 2 diagnostics-13-00018-t002:** Baseline information.

	High-Volume Laxative (*n* = 83)	Very Low-Volume Laxative (*n* = 83)	*p*-Value
**Age**, years, median (range)	51 (20–87)	55 (19–87)	0.28
**Male sex** (%)	43 (52)	38 (46)	0.44
**BMI**, median (range)	25.2 (16.0–42.8)	25.4 (16.2–41.0)	0.96
**Colonoscopy prior to CCE** (%)	18 (22)	25 (30)	0.21
- Incomplete	13	18	
- Refused	5	7	
**Indication for CCE**			0.37
IBD (%)	43 (52)	35 (42)	
Known IBD: - Crohn’s disease	34	30	
	25	22	
- Ulcerative colitis	3	5	
- Indeterminate colitis	6	3	
Suspected IBD	9	5	
GI bleeding (%)	23 (28)	31 (37)	
- Overt	14	15	
- Occult	9	16	
Other (%)	17 (20)	17 (20)	
- Abdominal pain	12	8	
- Colon cancer screening	5	5	
- Weight loss	0	2	
- Diverticulitis	0	2	

**Table 3 diagnostics-13-00018-t003:** Capsule endoscopy performance.

	High-Volume Laxative *n* (%)	Very Low-Volume Laxative *n* (%)	*p*-Value
Complete examinations	64 (77)	60 (72)	0.48
Adequate cleansing grade SB	73 (88)	75 (93)^a^	0.32
Adequate cleansing grade colon	56 (67)	59 (75)^b^	0.31
Complete examinations with adequate cleansing grade—SB	57 (69)	59 (71)	0.74
Complete examinations with adequate cleansing grade—colon	45 (54)	52 (63)	0.27
Capsule retention in the small bowel or stomach	9 (11)	11 (13)	0.63
Transient	9	9	
Permanent	0	2	
Transit time, minutes, median (range)	370 (78–957)	467 (110–1128)	0.25
Stomach	37 (3–296)	85 (1–498)	<0.01
Small bowel	78 (11–565)	79 (21–847) ^a^	0.77
Colon	210 (7–890)	244 (11–760) ^b^	0.89

^a^ out of 81 as two were missing; ^b^ out of 79 as four were missing.

**Table 4 diagnostics-13-00018-t004:** Capsule endoscopy outcome.

		High Volume Laxative *n* = 83 (%)	Very Low Volume Laxative *n* = 83 (%)	*p*-Value
Colon	Normal	34 (41)	28 (34)	
	Positive findings	49 (59)	55 (66)	0.34
	Findings ^a^	63 (NA)	60 (NA)	
	- IBD	15	16	
	- Polyps	17	14	
	- Cancer	1	1	
	- Angioectasias	5	6	
	- Diverticulosis	22	17	
	- Hemorrhoids	2	5	
	- Other	1	1	
Small bowel	Normal	51 (61)	57 (70 ^b^)	
	Positive findings	32 (39)	24 (30 ^b^)	0.23
	Findings	32 (39)	24 (30 ^b^)	
	- IBD	16	11	
	- Polyps	1	2	
	- Angioectasias	13	10	
	- Other	2	1	

^a^ more than one finding per patient possible; ^b^ out of 81 as two were missing.

## Data Availability

The data sets are not publicly available due to the clinical nature of the data but are available from the corresponding author on reasonable request.

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
