# Peer review of "The Effectiveness of a Very Low-Volume Compared to High-Volume Laxative in Colon Capsule Endoscopy"

_diagnostics, 2022, doi:10.3390/diagnostics13010018_

Round 1

Reviewer 1 Report

It is a well designed paper, written in a logical manner and having clinical impact, in the way of improving the colon cleansing rate and subsequently the diagnostic accuracy of  colon capsule endoscopy. Moreover, the authors highlight the strengthens and limits of their study and also refer to future directions in order to improve the bowel preparation and the technique of colon capsule endoscopy. I would recommend authors to remove reference number 34 which mention a paper still under review.  

Author Response

Thank you very much for the opportunity to revise the manuscript.

Reviewer 2 Report

This is an interesting study that aims to compare low and high volume bowel prep regimens at panintestinal capsule endoscopy using the Pillcam Crohn’s capsule. Overall the study and its results are interesting and a useful addition to the literature. I have some minor comments: 

  1. The study is retrospective and single centred. One of the limitations as a result is the relatively small numbers of patients, and the inability to perform power calculations to be able to detect small differences in completion rates and bowel prep adequacy. Larger, multicentre, prospective studies would be needed to confirm these findings in the future. I think this can be simply mentioned in the discussion.   

2. The PillCam® Crohn’s capsule (PCC) was used rather than the colon capsule (CCE-2). I understand this was used for panintestinal investigation in some cases (eg assessment of small bowel and colonic Crohn’s disease) and as colonic investigation in other indications (eg colon screening). Please confirm in the study design section.     

3. Results: "We found a CR and colonic ACR of 77% and 67%, respectively, in the high-volume group and 72% and 75% in the very low-volume group. In the high-volume group, 54% had complete transit with adequate colon cleansing, whereas this was the case for 63% in the very low-volume group."

These results are rather disappointing, with just over half of the cases being successful (complete with adequate prep quality). A bit more discussion and comparison with other studies in the literature would be useful and whether this reflects the experience in other centres. 

4. The authors discuss the results from the Swedish study which showed that vomiting was  associated with the very low-volume regimen at colonoscopy, possibly due to delayed gastric emptying due to the presence of ascorbic acid in the bowel prep solution. I wonder if they had a similar experience in their capsule study, ie did patients in the low volume prep group experience more vomiting with this prep or were they able to tolerate it well despite the more prolonged gastric emptying time? 
